# Determination of Population Mobility Dynamics in Popayán-Colombia during the COVID-19 Pandemic Using Open Datasets

**DOI:** 10.3390/ijerph192214814

**Published:** 2022-11-10

**Authors:** Andrés Felipe Solis Pino, Ginna Andrea Ramirez Palechor, Yesid Ediver Anacona Mopan, Victoria E. Patiño-Arenas, Pablo H. Ruiz, Vanessa Agredo-Delgado, Alicia Mon

**Affiliations:** 1Facultad de Ingeniería, Corporación Universitaria Comfacauca-Unicomfacauca, Cl. 4 N. 8-30, Popayán 190001, Cauca, Colombia; 2Universidad Nacional de La Matanza, Buenos Aires B1754, Argentina

**Keywords:** COVID-19, population mobility, exposure indicator, movement range maps, Facebook data for good, mobile phone location data

## Abstract

The COVID-19 pandemic is a catastrophic event that marked the history of humanity. The virus’s transmissibility has primarily prevented the control of the pandemic, so it has become vital to determine and control the dynamics of the population mobility to reduce the epidemiological impact. Considering the above, this paper uses an exposure indicator based on the movement ranges provided by Facebook to determine the dynamics of population mobility in Popayán city for the period after the appearance of COVID-19. Using statistical analysis techniques, it then contrasts the data obtained with the public circulation reports provided by Google and Apple. The results suggest that the exposure indicator is reliable and presents moderate to strong linear relationships for the public data, which implies that it can be an additional resource for decision-making to curb the spread of the virus.

## 1. Introduction

The 2019 coronavirus disease, known as COVID-19, is one of the most catastrophic events in human history, with millions of deaths worldwide [1]. The consequences and impact of the pandemic on the social [2,3], economic [4], and mental health [5] levels in the human species have been severe. Specifically, in Colombia, these effects have been reflected in socioeconomic indicators such as work, poverty, inequality, and quality of life, which have had a major setback compared to the project for 2021 [6]. This has given rise to widespread discontent among the Colombian population besides some reforms proposed by the government, which has caused a multitude of people to take to the streets to demonstrate against the current situation of the country and the government’s proposals [7].

The consequences and changes that may result from the demonstrations in the country are yet to be seen, thanks to the meetings held by the government, with the different unions and minority groups to agree. Colombian authorities are concerned about the crowding of people in the demonstrations, as it is widely accepted that mass gatherings and migrations play an essential role in spreading contagious diseases [8]. Therefore, an increase in the number of COVID-19 infections is expected, leading to a potential fourth wave of infections that could cause the collapse of the Colombian health system [9]. It is essential to estimate the participation of people in demonstrations to pay special attention to zones (administrative areas) where massive concentrations occur and take measures against the spread of the virus in those places.

With the constant evolution of technologies, it has sought multiple options to respond to the different natural disasters that afflict humanity [10,11]. The scientific community’s proposal is related to crisis mapping, which involves collecting, visualizing, and analyzing large amounts of data in real time during a crisis, allowing more effective responses to the calamities occurring [12]. As a result, companies with large amounts of data have developed tools and resources that provide reliable, on-hand information to help solve problematic situations [13]. Specifically, for the COVID-19 pandemic, the multinational company Facebook, Inc. enabled a series of technological resources through high-resolution maps, which allow researchers to obtain large amounts of anonymized data to conduct their research; this initiative has been called “Facebook data for Good”. Likewise, Google, LLC has released anonymized data sectioned by administrative zones to estimate the displacement of populations; this initiative has been called “COVID-19 Community Mobility Reports”.

The above are examples of the actions taken by these companies to mitigate the impact of the pandemic [14]. This is the case of Mena et al. [15], who used high-resolution maps to find the disease’s incidence and mortality patterns and its dependence on the socioeconomic level in Santiago de Chile-Chile. The results suggest that people living at a low socioeconomic level present higher deaths from the infection, indicating that socioeconomic inequalities are a preponderant factor in COVID-19 mortality. Another research that uses demographic mobility maps to determine the impact of quarantine in the world is that of Kishore et al. [16]; in this paper, the authors state that after the quarantine closure announcements in the different countries, there was an abnormal displacement of people from the city to the countryside, for this an agent-based spatial model was used. The results suggest that the greater the population movement, the greater the likelihood of COVID-19 cases occurring in rural areas, so they conclude that blockades play an essential role in reducing virus infections and controlling outbreaks in the countryside.

Similarly, research by Zachreson et al. in [17] uses mobility data proposed by Facebook and Google to estimate potential transmission risks of asymptomatic COVID-19 individuals using mobile devices that detect prominent people’s movements in Australia and then compare the estimates with government data. The results suggest that movement data provided by mobile devices are a good predictor of geographic areas with high COVID-19 exposure. It is concluded that it is a reliable prediction method and could help implement better geographical restrictions in administrative areas. Finally, other works such as Heo et al. in [18] and Spelta et al. in [19] are samples of how mobility data are used to determine crowd movements and their impact on socioeconomic indicators, which speaks that this type of information is reliable for making different decisions in times of crisis. However, to the authors’ knowledge, no proposals have been made to estimate the rate of exposure to COVID-19 in Colombia during the 2021 pandemic and social protests, which could lead to new waves of COVID-19 infections. Considering the above, there is a clear need to implement tools to estimate the exposure of different population groups to take the corresponding precautions and prevent the spread of the disease.

Considering the above and seeking an effective and fast way to determine how the inhabitants of Popayán city have been exposed to COVID-19 in the period between 1 March 2020, and 19 May 2021, a proposal is presented to estimate the exposure to the virus of different population groups through high-resolution maps and datasets. This allows for obtaining a reliable Exposure Indicator (EI) based on mobility data, which is accepted as one of the main variables explaining the increase or not of COVID-19 transmissions and infections [20,21,22].

Therefore, the major contribution of this research is to expand the area of knowledge through empirical evidence of COVID-19 exposure in different population groups by using high-resolution maps and mobility data, generating an exposure index to provide a new source of information that will allow health authorities to take preventive actions in case of a new infectious disease or COVID-19 outbreak.

Finally, the rest of the paper is divided into the following sections: the second section presents the methodology and resources used, the third section summarizes and shows the results obtained, and the fourth section summarizes the main conclusions and future work.

## 2. Materials and Methods

This section presents the materials and methods used to obtain the results and the statistical techniques employed for data analysis. The GADM global administrative areas database [23] was used to locate the mobility data. High-resolution maps and commuting data provided by Facebook [24], mobility reports published by Google [25], and movement data provided by Apple [26] were also used. With the shower resources, the exposure of different population groups in Popayán city (men, women, and young people (between 15 and 24 years old) to national protests in Colombia in the year 2021 were estimated and analyzed, and compared with a period of lower mobility such as the COVID-19 pandemic. In addition, QGIS software [27] was used for map manipulation and geographic calculations, while Paleontological Statistics Software (PAST) [28] was used for statistical treatments. Subsequently, with the data obtained, a descriptive analysis, besides the Shapiro-Wilk test [29], was executed to check the data’s normality, resulting in a non-normal distribution. Finally, Spearman correlation tests [30] were used to determine the dependence between the study variables (exposure indicator and data supplied by Google and Apple).

### 2.1. Construction of Movement Range Maps by Administrative Areas with Its Exposure Indicator

For the construction of the movement range maps, different sources of information provided by Facebook are used; this information is collected from users with mobile devices with the location history activated or using some of the company’s services. It is estimated that in 2021 almost 2 billion people will use the Facebook mobile application [31], so this is an essential group of citizens that can characterize a large part of the population.

The maps built for this research are composed of three main components: the first component is given by the movement trend data that are calculated based on two metrics managed by the company, one is the change in the movement of populations, and the other is the permanence in administrative areas during the day. These metrics provide insight into the movement of different population groups in administrative areas worldwide. Specifically, change in movement refers to the number of people moving from one place to another and compares this to a baseline period prior to the COVID-19 restriction measures; movement is considered when a person moves from one point to another within 600 m. The permanence metric determines what swath of the population remains stationary in administrative areas throughout the day [24]. Movement change data has been used in multiple investigations and is accepted as a reliable technological resource for determining population mass movements [32]. Likewise, the second component that allows the construction of maps is provided by GADM, which is a compendium of maps and spatial data of all countries and their subdivisions. This resource allows for the spatial localization of the movement trend data presented above. Finally, the third component refers to the data sets related to population density, which allows the calculation of population values for each administrative zone within the study area (Popayán city).

The commuting range maps merged with the maps provided by GADM and the Facebook datasets (Figure 1). It combined the demographic statistics provided by the population density datasets and the population movement metrics. These datasets made it possible to construct the indicator to determine the administrative areas with the highest exposure to COVID-19 based on mobility data at different time-lapses. The methodology followed to determine the exposure indicator at specific times. First, QGIS was used to combine the movement and population density datasets into a temporal layer, then a time window (in days) was selected to filter the movement data, and it merged the constructed temporal layer with the GADM administrative polygons. Subsequently, the population values of each administrative zone were computed, and the percentages of each population group were calculated. The above data calculated the weighted EI based on the ratio between the percentage of the population group and the percentage of people remaining in their homes or surroundings, all weighted by the total population group in the zone (Equation (Equation 1)). Finally, each of the values found for each administrative zone is scaled from 0 to 100 and is symbolized on the maps provided by the GADM. Next, the equation for determining the exposure indicator based on the work presented by Herdağdelen in [33] is presented.
(1)IE=((GPM∗100)∗(100−(PI∗100)))∗TGP
where *GPM* is the percentage of people in the population group, *PI* is the percentage of immobile people, and *TGP* is the total weighted sum of the population group.

Figure 2 shows a map generated with the methodology described above, presenting different Colombian municipalities with the obtained exposure indicator. It is essential to mention that there are administrative areas that are not colored and do not present an indicator; this is because there are municipalities in which the number of users that provide location data to Facebook is not sufficient, so it was not possible to produce movement range data in a way that preserves user privacy and data quality.

### 2.2. Estimation of Changes in Local Mobility

For the evaluation of changes in the exposure indicator according to the mobility ranges provided by Facebook, data provided by Google were used to compare these metrics with those obtained from the maps. The local mobility reports provided should help public health authorities understand the Changes in Mobility (CM) in different areas (workplaces, parks, food stores, and residential areas, among others) to take positions in favor of reducing the spread of the virus. Unlike the datasets provided by Facebook, these reports do not report a specific number related to mobility. However, starting from a reference value, they show by what percentage it has varied for the period studied, which allows observing mobility trends by region category and location. The baseline value is computed from a time from 3 January to 6 February 2020, which is a time prior to the general disruption and the response of communities to COVID-19 in each country; this allows to establish a starting point for analyzing how the dynamics of mobility changed because of the mandatory quarantine measures. Mobility data is computed from the location history of users using Google services. Finally, it is essential to note that all the information provided is reliable data that meets a high standard of quality and privacy [34], so government authorities and researchers can use it.

Another company that provides data on people’s mobility based on the location of the different devices it manages is Apple Inc, which does not detail how it executes the data preprocessing but assures that it is anonymous and reliable. Unlike previous datasets, Apple uses as a reference only 13 January 2020, to establish a point of analysis of mobility data during the COVID-19 outbreak and thus interpret how it has changed Mobility Trends (MT) in the population through percentages based on a reference value [26], in addition, it should be noted that these data are not segmented into administrative areas, but are general reports for Colombia. Finally, it should be mentioned that Apple segments the MT data in populations that move by driving or walking.

## 3. Results and Discussion

This section presents the results obtained in this research. First, the COVID-19 exposure indicator for different population groups is determined and presented between 1 March 2020, and 19 May 2021, framed by two critical events, such as the COVID-19 pandemic and the social demonstrations of 2021. Then, the EI obtained for different administrative areas of Colombia was compared to find differences between the dynamics of population displacement. Finally, the data obtained (exposure indicator for Popayán city) is contrasted with the information provided by Apple and Google, using Spearman correlation coefficient tests to find the degree of linear association between the different sources of information.

### 3.1. Exposure Indicator Based on Mobility Data for Different Periods in Popayán City

Next, using the methodology expressed in the previous section, the exposure indicator was determined for the population group of men in the municipality of Popayán for the period between 1 March 2020, and 19 May 2021. The choice of this administrative area is because it is a municipality that has been the epicenter of different mass demonstrations during the protests in Colombia in the year 2021; this allows finding significant variations in the exposure index based on mobility and comparing it with periods of lower population displacements, such as those of the mandatory quarantine [35]. It is important to note that the EI was determined daily because of the frequency of updating mobility data by Facebook, Google, and Apple.

The exposure indicators obtained were sectioned into different periods to facilitate the presentation and understanding of this information; important events frame the division during the COVID-19 pandemic [36]. Between 1 March 2020, and 5 March 2020, which was termed the “information stage”, there were no reports of COVID-19 infection in Colombia. Then, there was a period between 6 and 23 March 2020 (called “pre-isolation”) in which the first infections by the virus were reported, applying some restrictions in gatherings of over 500 people and declaring a sanitary emergency. Then, from 24 March to 30 August 2020, a mandatory national quarantine was established, in which people could not leave their homes except in cases of first necessity, which presumably decimated the mobility of the population. Finally, from 1 September 2020, to the present, a new quarantine phase called “selective isolation” was started, which comprised the gradual reopening of different public establishments and the partial return to normality in the country. However, it is essential to show that the period between 28 April 2021, and 19 May 2021, was also sectioned. Massive protests and mobilizations occurred in the country within the context of the 2021 Colombian protests.

Figure 3 shows the exposure indicators for the male population group between 1 March 2020, and 19 May 2021, which corresponds to the COVID-19 quarantine period and the social protests of 2021. It can be shown that as time has passed, the population has taken fewer precautions in the trips they make; It reflected this in the exposure indicators studied, where it is observed that there is an upward trend line with an R-Squared of 0.366, which speaks of a general relaxation of the population in this aspect.

Specifically, it can be observed that the first five days corresponding to the informative stage had the highest exposure values in the population (on average, 28.2). This allows inferring that during the first days of March in Popayán, there was a lack of knowledge about the forms of transmission of the virus, and the population did not take precautions to remain in their homes. This has to do with the fact that positive cases of COVID-19 had not yet been identified in Colombia at this stage, so there was some sense of remoteness with the virus. However, the average exposure indicator in the pre-isolation stage decreased by 4 points (on average 24.1) concerning the information stage; this may be because the first positive cases of the virus were reported, and the government took measures against mass gatherings of people, although without reaching the level of mandatory quarantine. Specifically, the most significant drop in EI in this phase occurred between 21 and 22 March 2020 (from 18 to 10), which coincides with the first death caused by the coronavirus in the country, which may have generated some fear in the population resulting in a lower movement of people. Subsequently, during the mandatory national quarantine stage (Figure 4), the population’s exposure decreased by almost four points compared to the pre-isolation phase, suggesting that the measures adopted by the national government to curb the spread of the virus affected the mobility of the population. However, the population increased its movements over time at the beginning of the quarantine. This is clear when analyzing the population indicator month by month or by stage (Table 1); for example, between 24 March and 24 April 2020, the average was 12.6, and between 25 April and 24 May 2020, the average increased to 15.9, in the following month (between 25 May and 24 June 2020) the average was 19.9, while for the subsequent month (25 June to 24 July 2020) it was 21.8, finally for the corresponding month between 25 July and 24 August 2020, the average exposure indicator was 22.2. This implies that as the mandatory quarantine progresses, people take more minor precautions in their travels, ignoring the rules promulgated by the national government.

In the selective isolation stage, there was an increase in the number of people in the streets with an average of 24.9, representing an increase of almost 6 points in the exposure indicator, reflecting that people moved more freely as the stage changed. In the period between 31 August to 31 December 2020, Popayán noticed an exciting pattern on weekends, especially on Sundays. On these days, one would think that the population would move to a greater extent and the index would increase due to work breaks, but the opposite occurred since, on this day, people, on average, moved two points less than on the other days of the week. It also performed a specific analysis on dates in which, presumably, there were large influxes of the public in the Popayán city; this is the case of 31 August 2020, which was the first day of selective isolation in Colombia, where a score of 24 was obtained, which is not an enormous change for the trend of previous weeks. The same occurred on 21 September (labor rights protest) and 31 October (Halloween), which were dates where large mobilizations of people were presumed, but this was not reflected in the exposure indicator presenting values of 24 and 25, respectively, which are in line with what was reported for the previous days. However, a date that showed a differential increase in exposure was 24 December 2020, where the calculated indicator was 28. This value is one of the highest in the period analyzed, maybe because the population purchases for these festivities on that day and trips to social gatherings. Finally, in Popayán, there was a dramatic decrease in the EI on the third and fourth weekends of January 2021, with values similar to those of the national mandatory quarantine. This has to do with the fact that, for these dates, a decree was declared, enacting a mandatory curfew for non-essential people during these weekends [37], reducing the mobility of people in the city.

Finally, for the period of social protests in 2021, the exposure indicator decreased by almost 1.5 over the stage of selective isolation; this is explained because, although the protests against the measures proposed by the government have been massive, a large part of the laboring mass that went to work on a typical day has stayed at home to avoid confrontations with the public forces, which significantly reduces the displacement of the population on protest days, which is reflected in the exposure index. According to (Figure 5), it can be observed that in the first week of the 2021 Colombian protests, there was greater exposure of people because of more significant population displacements with an average value of 24.1, while for the second and third weeks, the average value was 23.5 and 22.8 respectively, which shows that there was a significant decrease in the number of people who mobilized. This is also clear when drawing a trend line for this period, where it can be observed that it has a low propensity with an R-Squared of 0.213. This suggests that as the Colombian protests passed, the population mobilized less, staying at home, which could be caused by multiple factors such as some agreements with the national government or the lack of food and supplies in Popayán. Now, a specific analysis should be made on 13 May 2021, where the EI was 25, being an abnormal value for the following days; It could have caused this because the population was in a state of shock because of the alleged case of sexual abuse of a young girl from Popayán, which unleashed the generalized anger of the population, calling for massive protests in the city.

### 3.2. Comparison between the Different Population Groups of the Popayán City According to Their Exposure Indicator

It analyzed the exposure indicator obtained with the different population groups of Popayán in the period between 1 March 2020, and 19 May 2021. The population groups selected were men (regardless of age), women (regardless of age), and young people between 15 and 24 years old (regardless of gender). It chose these population groups to compare the exposure to COVID-19 between men and women in different social contexts, such as the Coronavirus pandemic and the social protests of 2021. Likewise, the population group of young people was selected because, presumably, they are the ones with the highest participation in the protests, so it could be observed if the change in the EI is significant concerning this period.

Figure 6 shows a comparison between the different population groups described using a line diagram, where it can be seen that women and men have practically the same patterns of exposure to COVID-19 for the period analyzed, with similar indicators. Both population groups present ascending trend lines with close R-Squared values (R^2^ for women 0.374 and R^2^ for men 0.366), so it can be presumed that exposure to the virus could progressively increase because of the greater mobility of the population groups. Likewise, for the youth population group, the dynamics of exposure to the virus are like that of other population groups, with higher values in the information stage and significant decreases in the quarantine stage. In the period of social protests, an increase in exposure was presumed because of the greater mobility of young people, but this was not reflected in the indicator, although a significant increase was observed for 28, 29, and 30 April 2021, where participation was higher than for other days of protests. Regardless, it is assumed that the exposure indicator is lower for the youth population group because it is based on a smaller sample size than the population groups of women and men. The published data considers young people between 15 and 24 years of age, while for the population group of men and women, people 13 years of age and older represent a much more extensive population range.

Likewise, Spearman correlation coefficient tests were performed between the different population groups for the period studied, where it was found that they present strong positive correlation values between each other, especially for the group of men and women (ρ = 0.97); likewise, for the youth population group, the coefficients drop a little, but are still significant and with strong relationships between them (for men ρ = 0.92 and women ρ = 0.91). The above implies that beyond the population group studied, the dynamics of mobility of people and exposure to the virus are similar in the Popayán city for the period studied.

### 3.3. Comparison of Different Administrative Areas According to the Exposure Indicator

This section compares and analyzes the exposure indicator obtained among 25 administrative areas (municipalities) of Colombia between 1 March 2020, and 19 May 2021, some with similar sociodemographic characteristics and populations, emphasizing the municipality of Popayán. For this purpose, the exposure data based on the mobility of each municipality are organized and compared, and their Spearman correlation coefficient is determined for each administrative area. The correlogram matrix produced (Figure 7) is based on 445 observations. It displayed positive correlations in blue and negative correlations in red.

In general, it can be observed that the correlation coefficients found for 20 of the 25 municipalities studied are positive and can be moderate to strong and statistically significant. Although, it should be noted that there are some exceptions in the municipality of El Tambo (Cauca department), Uribia (La Guajira Department), El Paso (Cesar Department), and Yolombó (Antioquia Department) that presented low correlation values concerning other administrative areas. This allows inferring that the dynamics of mobility and exposure among the different municipalities were similar, with a decrease in the population’s movement under strict quarantine and a progressive increase in selective isolation.

However, some municipalities (El Tambo, Uribia, El Paso, and Yolombó) presented low correlation coefficients, which can be explained because a large part of the population of these municipalities is in rural areas (approximately 80%). This behavior means that most of the population does not act as a source of mobility data for Facebook because, in the countryside, there are difficulties in accessing the internet (it is estimated that in Colombia, internet access in the countryside is less than 10%) [38], which could lead to unrepresentative data. This can be corroborated when analyzing the municipality of La Virginia (Risaralda Department), which presents strong correlation coefficients (approximately 0.9) and has a population of close to 27.000 people, of which only 2.3% is rural. This implies that the total number of people in an administrative area is not an excluding factor for presenting low correlation coefficients for other municipalities, while internet access could be a determining element.

Finally, with the information presented in the correlation matrix, it can be shown that the strategy implemented by the Colombian national government to curb the expansion of COVID-19 was followed in the 25 municipalities analyzed since they presented similar dynamics of mobility and exposure throughout the study period except to a lesser extent for the period of social protests. However, if the analysis focuses on the 2021 protests, the EI varies in the municipalities, with correlation values no more significant than ρ = 0.82. This allows inferring that each municipality has had different mobility and exposure dynamics for this period and that it has thoroughly followed not all demonstrations called at the national level. This does not mean that the population did not show in these administrative areas, but they did so at different times (days), unlike the quarantine period in which the call to stay at home was generalized and followed by a large part of the population of these municipalities.

### 3.4. Comparison between the Exposure Indicator and Mobility Data from Google and Apple

This section confronts the exposure indicator based on movement ranges with the CM reports provided by Google and the MT provided by Apple. The data is based on information collected by mobile devices using the services of the companies mentioned above. This section does not compare whether one data source is more accurate than another, as this would not be possible because they use different user bases, and the processing and segmentation of the data are different for each source. However, what can be analyzed is whether there are similar trends in the dynamics of population movements for the study period. Spearman’s correlation coefficient was used to compare the relationship between these variables from 1 March 2020, to 19 May 2021.

The correlogram produced (Figure 8) is based on 445 observations. It displayed positive correlations in blue and negative correlations in red. The intensity of the color is proportional to the correlation coefficient, and the color legend shows the correlation coefficients and the corresponding colors. A negative correlation implies that its two variables are considered to vary in opposite directions. A positive correlation implies that its two variables vary in the same direction.

In Figure 8, the results suggest moderate to strong positive correlation coefficients (between ρ = 0.57 and ρ = 0.95) in the study variables (exposure indicator, Google’s CM, and Apple’s MT). The above implies that the exposure indicator developed in this paper is reliable and that its data based on population displacements can serve as a resource to study the dynamics of population mobility in administrative areas. Specifically, if the EI values are confronted with the data provided by Google reports, high positive and negative correlation coefficients are found in the population groups studied (men, women, and youth). In the reports segregated by Google locations (between ρ = 0. 76 and ρ = 0.95 for the positives and between ρ = −0.84 and ρ = −0.91 for the negatives), this may be because, in both companies, a similar reference period was used to observe and analyze the changes in mobilizations because of the quarantine and selective isolation period.

Now, if the analysis focuses on the variable “CM in residential areas” (Figure 9), it can be observed that this has strong negative correlation coefficients concerning the other variables. This section exposes the change in the duration that the population spends in their homes or surroundings, while in the rest of the categories (stores and leisure, supermarkets and pharmacy, workplaces and parks), the change in the total number of visitors is reflected [39]. The above validates what was found in the exposure indicators. For the quarantine period, this decreased drastically because the movement of people was reduced, while for the residential areas, there was a notable increase because people spent more time (duration) in their homes or surroundings because of government measures. Likewise, it should be shown that the trend in residential areas increased on 17 and 24 January 2021, when the curfew was declared in Popayán city, as opposed to what happened with the exposure indicator, which decreased in those periods because the mobility of the population was reduced. Finally, it is noteworthy that the percentage of change in mobility increased if it analyzes the period of protests in 2021, where an upward trend is observed. This ratifies what was mentioned in previous sections where part of the population that moved regularly stays at home during this stage (also reflected in other categories such as stores and leisure, supermarkets and pharmacies, parks and workplaces where a downward mobility change is observed for this period).

Now, regarding the data provided by Apple (Figure 10), it can be observed that they are abruptly for the study period; they have moderate correlation values (between ρ = 0.58 and ρ = 0.60) for the exposure indicators determined for the same period. This drop in correlation coefficient may be because Apple used only one reference day (13 January 2020) to obtain the mobility trends, implying fewer reference data concerning the datasets provided by Facebook and Google. Another reason may be that Apple’s data is not segmented into administrative areas but provides values for Colombia, which may mean that they do not fully characterize the population of Popayán. The data provided by Apple reflect the mobility trends seen in the previous datasets, except for an increase in the first days of July 2020 reflected neither in the exposure indicator nor in Google’s CM percentage, so here we find a difference between the data obtained and those studied.

Finally, to replicate and contrast the research results, the resources and materials are made available through maps and population mobility datasets in the following link https://cutt.ly/4nJKfNW (accessed on 15 January 2021).

## 4. Conclusions

This paper exposes the dynamics of population mobility in Popayán city during the period of COVID-19 and social protests in Colombia in the year 2021. In principle, an indicator of exposure to COVID-19 based on a range of motion data (provided by Facebook) is used and confronted with data published by Google and Apple. Along these lines, it can be concluded that it strongly linked the virus exposure data to the mobility data provided by different sources, showing that something is closely related to variables in the pandemic context. Likewise, with the data and analysis obtained, it can be indicated that the mobility of the population of the city of Popayán decreased drastically during the period of mandatory quarantine, by almost ten percentage points concerning the reference period and increased progressively over time during the quarantine phase, which speaks of a certain relaxation of the population because of displacements and social distancing measures. Along these lines, for the 2021 social protest period, the data suggest slight increases (of almost one percentage point) in people’s mobilizations, but it did not sustain them over time.

Regarding the mobility data published by different sources, they are valuable data that can play an essential role in the management and control of the COVID-19 pandemic because they allow estimating the dynamics of population mobility, which could facilitate and guide decision-making about the virus.

Among the main future work that can be derived from this research is the comparison of the data obtained on the variation of COVID-19 infections and deaths to determine the relationship of the information with the impact of the virus on health.

## Figures and Tables

**Figure 1 ijerph-19-14814-f001:**
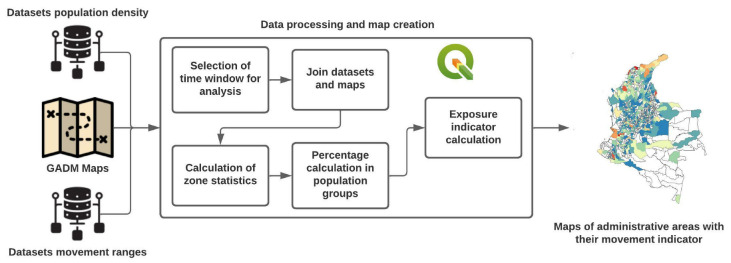
Process for constructing and determining the exposure indicator for each administrative area in the movement range maps.

**Figure 2 ijerph-19-14814-f002:**
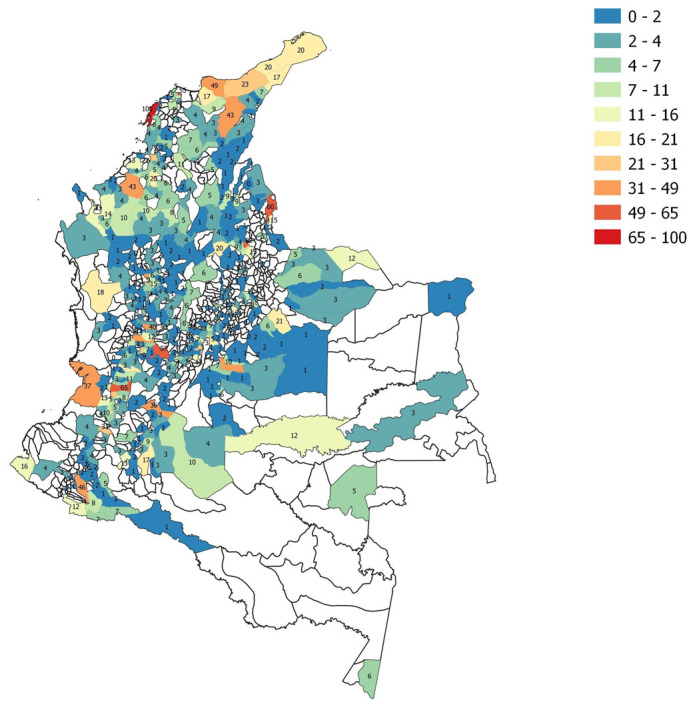
Example of range of movement map obtained according to the methodology described for 4 May 2021, with the exposure indicator in each administrative area.

**Figure 3 ijerph-19-14814-f003:**
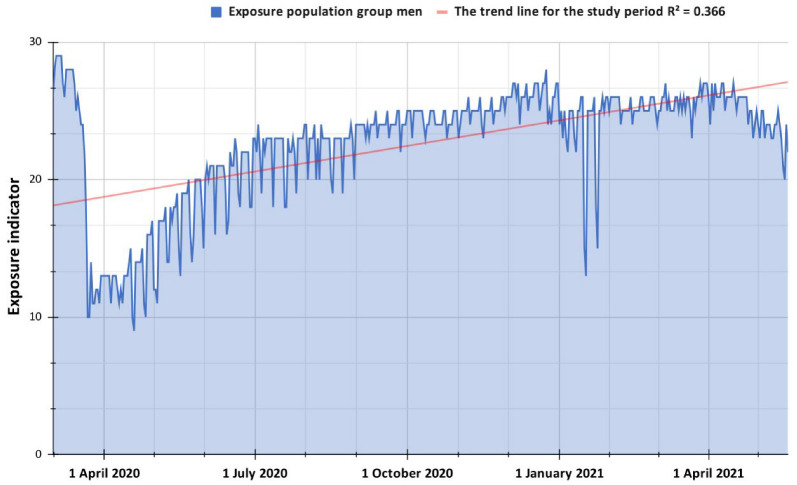
Exposure indicator based on movement ranges (Facebook) for 1 March 2020, to 19 May 2021, in the Popayán city.

**Figure 4 ijerph-19-14814-f004:**
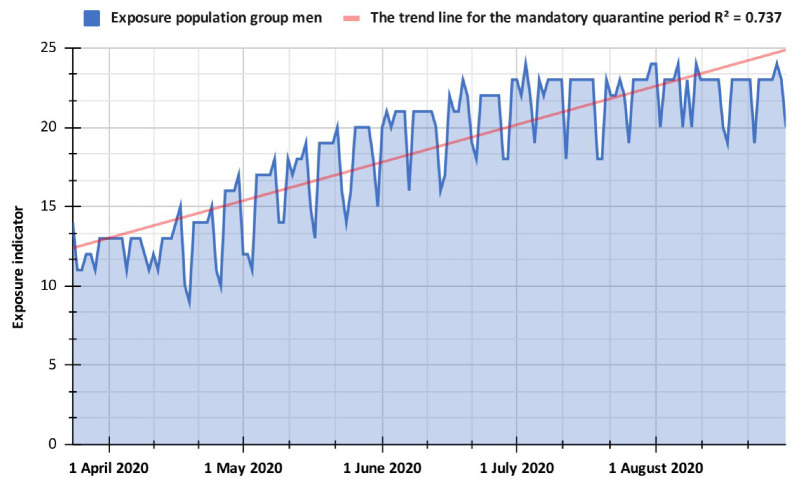
Exposure indicator for the mandatory quarantine period in the Popayán city.

**Figure 5 ijerph-19-14814-f005:**
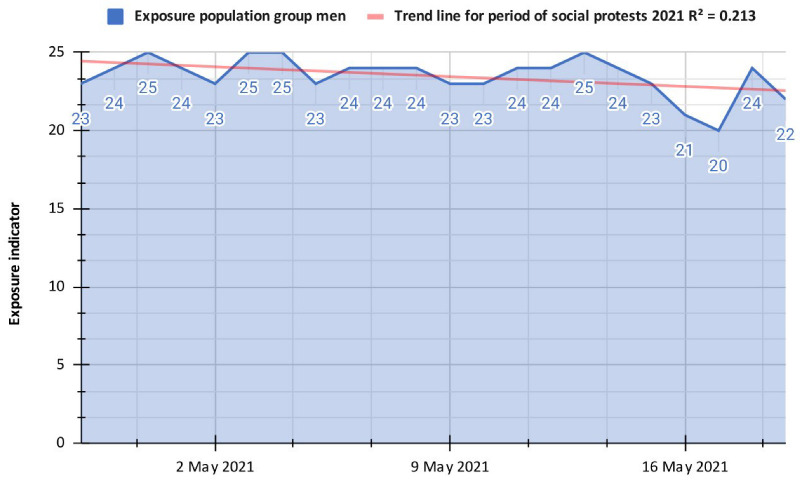
Exposure indicator for the period of social protests in 2021 in Popayán.

**Figure 6 ijerph-19-14814-f006:**
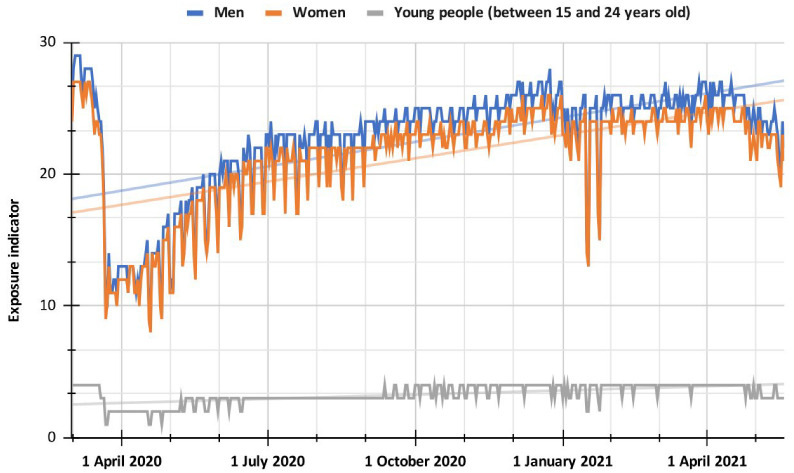
Comparison between the different population groups from 1 March 2020, to 19 May 2021.

**Figure 7 ijerph-19-14814-f007:**
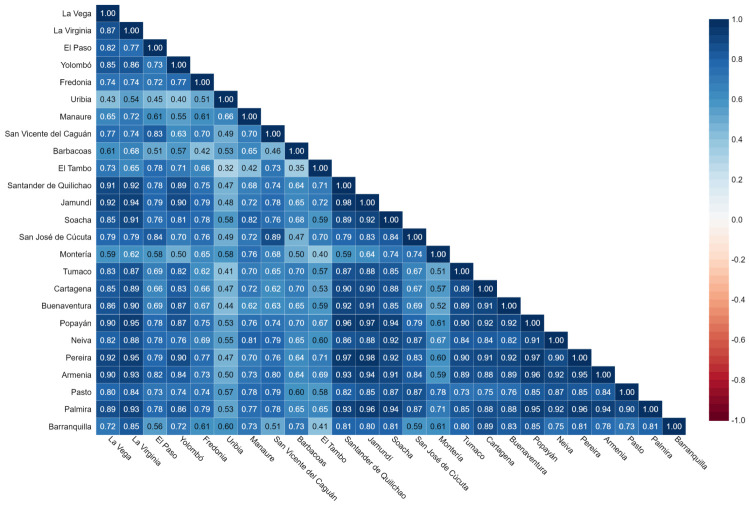
The correlation matrix reflects the linear relationship between 25 municipalities in Colombia for the period from 1 March 2020, to 19 May 2021.

**Figure 8 ijerph-19-14814-f008:**
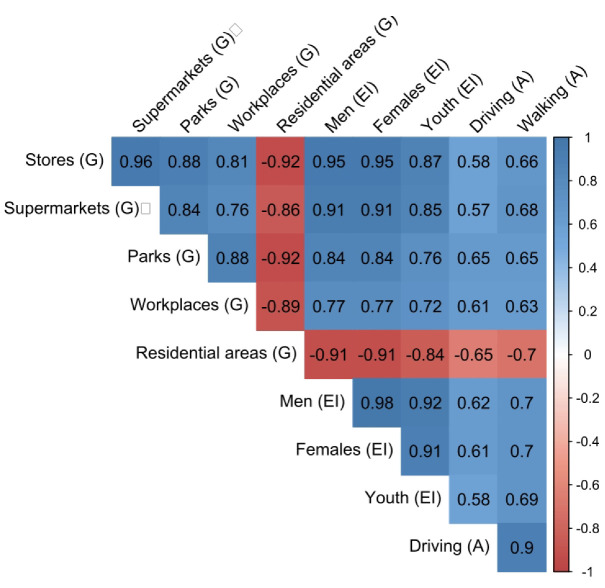
The linear relationship between the exposure indicator (EI) and the mobility reports provided by Google (G) and Apple (A) for 1 March 2020, and 19 May 2021.

**Figure 9 ijerph-19-14814-f009:**
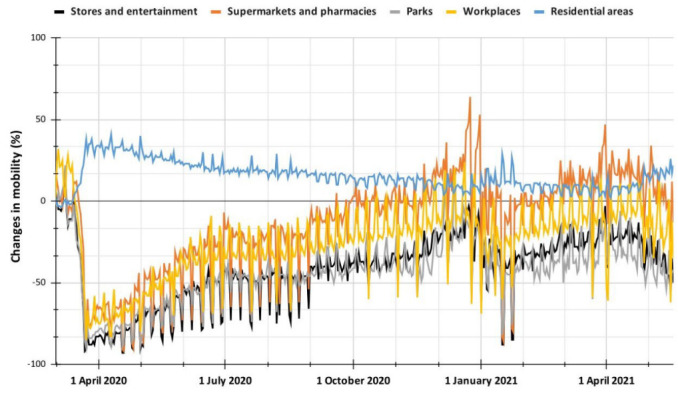
Changes in mobility as reported by Google for the period from 1 March 2020, to 19 May 2021.

**Figure 10 ijerph-19-14814-f010:**
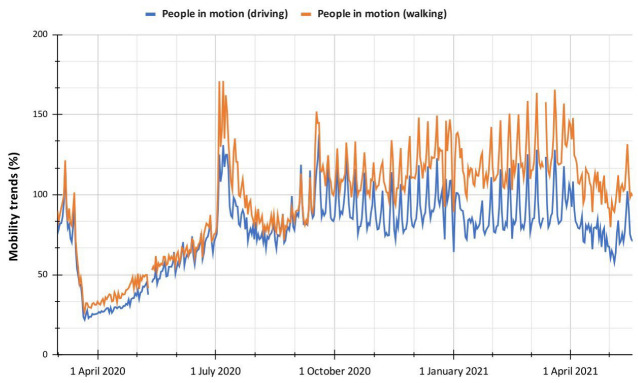
Mobility trends reported by Apple for 1 March 2020, to 19 May 2021.

**Table 1 ijerph-19-14814-t001:** Detailed information on the exposure ratio and its relation to relevant periods for the period under study.

Phase	Date	Average Exposure Indicator
Information stage	01/03/2020 to 05/03/2020	28.2
Pre-isolation stage	03/06/2020 to 03/23/2020	24.1
Mandatory quarantine stage	03/24/2020 to 08/30/2020	18.6
Selective isolation stage	08/31/2020 to 04/27/2021	24.9
Social protests in 2021	04/28/2021 to 05/19/2021	23.5

## Data Availability

To replicate and contrast the research results, the resources and materials are made available through maps and population mobility datasets in the following link https://cutt.ly/4nJKfNW(accessed on 15 January 2021).

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
