# Peer review of "Determination of Population Mobility Dynamics in Popayán-Colombia during the COVID-19 Pandemic Using Open Datasets"

_ijerph, 2022, doi:10.3390/ijerph192214814_

Round 1

Reviewer 1 Report

The authors used mobile data information provided by Facebook to determine population movement dynamics in the city of Popayan, Colombia, after the COVID-19 pandemic, comparing the data obtained with public reports published by Google and Apple. This study has important implications for the management and control of the COVID-19 epidemic. However, I have some suggestions for this manuscript, which are as follows:

1. The manuscript needs to change some details, such as punctuation: Page 2, line 71 "(datasets, This..." This sentence lacks a close bracket, etc.

2. Page 9 in result.section, 3.2. Comparison between the different population groups of the Popayan city according to their exposure indicator, the trend of COVID-19 exposure indicators for females and males is similar from Figure 6. Why is the exposure indicator of young population lower than that of males and females alone, and the trend range is small? Please explain.

3. Page 9, line 289, please check and confirm whether the "IE" in this sentence is correct. Do you want to express EI: Exposure Indicator?

4. Page 10, Line 326~332: About 80 percent of the population of El Tambo, Uribia, El Paso and Yolombo live in rural areas where access to the Internet is difficult. The data for the study was based on information collected from mobile devices served by the company, and this suggested that this population was not the source of Facebook's mobile data set. Does that make a difference in comparing Exposure Indicator in different populations? Please explain.

5. Please check the references section carefully. The format does not show consistency. Authors are requested to follow the journal's manuscript format, including the format of references and punctuation. For example, there are two periods at the end of sentences in literature 4, 5, 16, 17, etc. The year fonts of references 7, 10, 33 and 37 are not in bold.

Author Response

Popayán, Cauca.

8-octubre-2022

Dear reviewers

Subject: Responses to the corrections made

Cordial greetings; I now attach the corrections made to the article "Determination of Population Mobility Dynamics in Popayan-Colombia During the Covid-19 Pandemic Using Open Datasets". In general, it can be mentioned that the entire article was improved grammatically, the bibliographic references were updated, and corrections were made concerning the author's guidelines.

Reviewer/Editor

Reviewer's Suggestions

Author's Corrections

Review Report (Reviewer 1)

The manuscript needs to change some details, such as punctuation: Page 2, line 71 "(datasets, This..." This sentence lacks a close bracket, etc.

Accepted: The changes indicated by the reviewer were made, and the opportunity was also taken to rewrite phrases that could sound unformal, as well as the passive voice that could make reading difficult. The article underwent a thorough grammatical revision, with many improvements.

Page 9 in result.section, 3.2. Comparison between the different population groups of the Popayan city according to their exposure indicator, the trend of COVID-19 exposure indicators for females and males is similar from Figure 6. Why is the exposure indicator of young population lower than that of males and females alone, and the trend range is small? Please explain.

Accepted: The decrease in the exposure rate for young people has been explained. This is because it is a much smaller population group than the other groups. This means that there are fewer data to compare.

Page 9, line 289, please check and confirm whether the "IE" in this sentence is correct. Do you want to express EI: Exposure Indicator?

Accepted: we wanted to refer to the exposure indicator. IE was corrected to EI, and an extensive review and correction of acronyms were made to avoid any other errors.

5. Please check the references section carefully. The format does not show consistency. Authors are requested to follow the journal's manuscript format, including the format of references and punctuation. For example, there are two periods at the end of sentences in literature 4, 5, 16, 17, etc. The year fonts of references 7, 10, 33 and 37 are not in bold.

Accepted: References are checked in the article and in the reference list. But it is important to indicate that there is no precise format for this journal:

"References should be numbered in order of appearance in the text (including table titles and figure legends) and listed individually at the end of the manuscript. It is recommended that references be prepared using a bibliography software package such as EndNote, ReferenceManager, or Zotero to avoid typos and duplicate references."

Reviewer 2 Report

I have some suggestion to improve the manuscript
1. Keywords should include some advanced and important words. Please improve them.
2. Please provide fresh citations and references for statements and data in Section 1(Introduction). Moreover, refine the introduction by showing the importance of this study, the necessity of doing this research, its contribution to the literature, its novelty, major findings, and major policy implications.
3. In addition, in the introduction, please explain more the generalization of your study. Why your study is important? Clearly explains the problems that you are going to solve in this study.
4. In the introduction, line 1, the statement "The expanding inhabitants, advancements in innovation, and financial growth all contribute to the need for more energy." is not professional. Please rewrite it.
5. Please cite more fresh published papers for the paragraph of the introduction
6. Please use passive form throughout the text. In addition, do a proofreading to fix any typos an grammar mistakes. The paper must clearly express its case, measured against the technical language of the field and the expected knowledge of the journal's readership. Therefore, attention must be paid to the clarity of expression and readability, such as sentence structure, jargon use, acronyms.
7. Literature review: It is better to add a paragraph to the end of this Section to clarify the literature gap that the paper tries to fill it in. Better to summarize the literature review in the table with necessary items. Cite current and relevant
references from well reputed journals.
8. Please clarify the reasons to choose the variables. Is there any clear theory?
9. The paper requires a section on the theoretical background.
10. In brief, the methodology is ambiguous, and I could not understand what methodology they have used to achieve the research goals. In other words, the methodology needs modification with proper arguments in a scientific way. Therefore, the authors are advised to draw the conceptual framework to help readers understand the required steps to conduct the paper.
11. The novelty and contribution adopted model should be highlighted. If possible, the author could add the conceptual framework to show the methodological procedure at a glance. The discussion could be enhanced with additional outcomes of /comparisons with other studies (more recent studies within 2019, 2020, 2021 or 2022). Are your results aligned with those of other studies in the field? Yes, no, why, Please discuss and explain with more recent papers.
12. Please add/improve the robustness checks to ensure the validation of empirical findings.
13. Conclusions are not adequate. Please add clear practical policy implications。
14. Authors are suggested to improve the conclusion section as well since it broadly handled and should be very concrete for the description of the results followed by the policy. How your study can be benefited for society?
15. My suggestion to the authors is the necessity of presenting a graphical conclusion or summary for easy understanding of the results.

Author Response

Popayán, Cauca.

8-octubre-2022

Dear reviewers

Subject: Responses to the corrections made

Cordial greetings; I now attach the corrections made to the article "Determination of Population Mobility Dynamics in Popayan-Colombia During the Covid-19 Pandemic Using Open Datasets". In general, it can be mentioned that the entire article was improved grammatically, the bibliographic references were updated, and corrections were made concerning the author's guidelines.

Review Report (Reviewer 2)

Keywords should include some advanced and important words. Please improve them.

Accepted: New, more technical keywords were added as suggested by the reviewer

Please provide fresh citations and references for statements and data in Section 1(Introduction). Moreover, refine the introduction by showing the importance of this study, the necessity of doing this research, its contribution to the literature, its novelty, major findings, and major policy implications

Accepted: the major contribution of the article was rewritten, demonstrating the importance of the article for the literature and its novelty.

In addition, in the introduction, please explain more the generalization of your study. Why your study is important? Clearly explains the problems that you are going to solve in this study

Accepted: We consider that the importance of our article is reflected when it is mentioned that "generating an exposure index to provide a new source of information that will allow health authorities to take preventive actions in case of a new infectious disease or Covid-19 outbreak”. Therefore, due to the urgency of the pandemic, any new tool that can help to reduce the number of cases is a significant contribution.

 In the introduction, line 1, the statement "The expanding inhabitants, advancements in innovation, and financial growth all contribute to the need for more energy." is not professional. Please rewrite

We did an exhaustive search for that phrase and did not find it, but the introduction was completely rewritten more professionally.

5. Please cite more fresh published papers for the paragraph of the introduction

New articles were taken and cited

Please use passive form throughout the text. In addition, do a proofreading to fix any typos an grammar mistakes. The paper must clearly express its case, measured against the technical language of the field and the expected knowledge of the journal's readership. Therefore, attention must be paid to the clarity of expression and readability, such as sentence structure, jargon use, acronyms.

Accepted: the entire article was improved grammatically

Literature review: It is better to add a paragraph to the end of this Section to clarify the literature gap that the paper tries to fill it in. Better to summarize the literature review in the table with necessary items. Cite current and relevant

references from well reputed journals.

Accepted: We rewrote the works related in the introduction and also added the need and the problems faced by the study.

Please clarify the reasons to choose the variables. Is there any clear theory?

We believe that this comment does not apply, since we do not have input variables to our study.

The paper requires a section on the theoretical background

We believe that it is not necessary to have a theoretical section, since the introduction reflects 7 related papers, which are in fact the most relevant in the area. Also, as you have seen in the journal, it is unusual to use this section for IJERPH.

In brief, the methodology is ambiguous, and I could not understand what methodology they have used to achieve the research goals. In other words, the methodology needs modification with proper arguments in a scientific way. Therefore, the authors are advised to draw the conceptual framework to help readers understand the required steps to conduct the paper.

Accepted: a figure was added in the methodology explaining the methodology used in the study.

The novelty and contribution adopted model should be highlighted. If possible, the author could add the conceptual framework to show the methodological procedure at a glance. The discussion could be enhanced with additional outcomes of /comparisons with other studies (more recent studies within 2019, 2020, 2021 or 2022). Are your results aligned with those of other studies in the field? Yes, no, why, Please discuss and explain with more recent papers.

Accepted: The novelty of the study has already been highlighted in the introduction section. Now, the exposure indicator for Covid-19 was not applied in Colombia, comparing the data obtained with something similar is not easy. Future work is expected to use the exposure index and relate it to COVID infections, but we believe that this study would be a different approach, and a more in-depth analysis would have to be performed.

Conclusions are not adequate. Please add clear practical policy implication.

Not accepted: We cannot add policy implications, as it is beyond the scope of our study. We believe that our findings conclude the objective of our work, which is the implementation of exposure indicators during the 2021 protests and pandemic. Our conclusions mention this and discuss how low mobility will be in these months. Public policies should come out of a new study that takes into account what was done and what can be done to improve.

Authors are suggested to improve the conclusion section as well since it broadly handled and should be very concrete for the description of the results followed by the policy. How your study can be benefited for society?

Accepted: The conclusions were further clarified and the results found.

My suggestion to the authors is the necessity of presenting a graphical conclusion or summary for easy understanding of the results

Not accepted: We did not consider it necessary to perform a graphical analysis since we have graphs in each section for this purpose. Even so, the conclusions were improved to make them simpler to understand and more aligned with our results.

Reviewer 3 Report

1. I suggest splitting the paragraph lines 33-57.

2. What was the reason for choosing the group aged 15-24 years?

3. What is the reason that the socio-economic differences and education were not observed, since they have a direct influence on the studied exodus?

4. Improve the resolution and size of the maps and graphs since the regions and range are illegible.

Author Response

Popayán, Cauca.

8-octubre-2022

Dear reviewers

Subject: Responses to the corrections made

Cordial greetings; I now attach the corrections made to the article "Determination of Population Mobility Dynamics in Popayan-Colombia During the Covid-19 Pandemic Using Open Datasets". In general, it can be mentioned that the entire article was improved grammatically, the bibliographic references were updated, and corrections were made concerning the author's guidelines.

Review Report (Reviewer 3)

I suggest splitting the paragraph lines 33-57.

Accepted: the indicated paragraphs were divided and the wording of the introduction was improved.

2. What was the reason for choosing the group aged 15-24 years?

The reason is that this is the age range for young people that Facebook considers, and this is how it delivers the data. Therefore, this is out of our hands and is at the discretion of the multinational.

3. What is the reason that the socio-economic differences and education were not observed, since they have a direct influence on the studied exodus?

Although we consider that socioeconomic and educational variables influence the exposure index, we believe that taking them into account would be outside the scope of our study. In this line, it is also important to point out that the data provided by Apple, Facebook, and Google are anonymized by algorithms created for these purposes, so it is challenging to conceptualize educational or economic variables correctly without correctly differentiating the population groups studied. Therefore, we believe a new study could be conducted to take these variables into account and analyze them based on multivariate statistics. 

4. Improve the resolution and size of the maps and graphs since the regions and range are illegible.

Accepted: Improved the resolution of the maps, correlation matrices and images of all graphs.

Reviewer 4 Report

I have a following question for this paper.

It would be good to analyze the time reliability of this method. How accurate it is to reflect the period of phase stage. Taking this prediction in the context below an example (Line 222-225). Considering the time of COVID incubation/treatment period, the news gets to the public and the decision making of movement, do you see the movement ranges delayed a certain time of period?

“Specifically, the most significant drop in EI in this

phase occurred between March 21 and 22, 2020 (from 18 to 10), which coincides with the

first death caused by the coronavirus in the country, which may have generated some

fear in the population resulting in a lower movement of people.”

Author Response

Popayán, Cauca.

8-octubre-2022

Dear reviewers

Subject: Responses to the corrections made

Cordial greetings; I now attach the corrections made to the article "Determination of Population Mobility Dynamics in Popayan-Colombia During the Covid-19 Pandemic Using Open Datasets". In general, it can be mentioned that the entire article was improved grammatically, the bibliographic references were updated, and corrections were made concerning the author's guidelines.

Review Report (Reviewer 4)

I have a following question for this paper.

It would be good to analyze the time reliability of this method. How accurate it is to reflect the period of phase stage. Taking this prediction in the context below an example (Line 222-225). Considering the time of COVID incubation/treatment period, the news gets to the public and the decision making of movement, do you see the movement ranges delayed a certain time of period?

“Specifically, the most significant drop in EI in this

phase occurred between March 21 and 22, 2020 (from 18 to 10), which coincides with the

first death caused by the coronavirus in the country, which may have generated some

fear in the population resulting in a lower movement of peopl

As authors of the article, we believe it is an excellent point to consider the temporal precision of the exposure index. However, it is essential to mention that since a large number of days were analyzed, approximately 400, it is difficult to analyze this temporal complexity due to the numerous periods related to the pandemic, social protests, and economic periods in the country. Therefore, we consider it good future work, but it should not be marked yet in our initial work since the idea is to find a tool for the authorities to make decisions, and in future work, we could measure the temporal reliability of the exposure index.